# Optimization of Sample Preparation for Detection of 10 Phthalates in Non-Alcoholic Beverages in Northern Vietnam

**DOI:** 10.3390/toxics6040069

**Published:** 2018-11-19

**Authors:** Thanh-Thien Tran-Lam, Yen Hai Dao, Duong Thanh Nguyen, Hoi Kim Ma, Trung Quoc Pham, Giang Truong Le

**Affiliations:** 1Institute of Chemistry, Vietnam Academy of Science and Technology (VAST), 18 Hoang Quoc Viet, Cau Giay, Hanoi 100000, Vietnam; thanhthien307@gmail.com (T.-T.T.-L.); dhy182@gmail.com (Y.H.D.); ntduong182@gmail.com (D.T.N.); phamquoctrung0811@gmail.com (T.Q.P.); 2University of Science, Vietnam National University HCMC, Linh Trung Ward, Thu Duc District, Ho Chi Minh City 720400, Vietnam; makimhoi@gmail.com

**Keywords:** phthalate, non-alcoholic beverages, liquid–liquid extraction, response surface methodology, GC-MS/MS

## Abstract

A novel method was developed for the sensitive, cheap and fast quantitation of 10 phthalates in non-alcoholic beverages by liquid–liquid extraction (LLE) combined with gas chromatography tandem mass spectrometry (GC-MS/MS). The best results were obtained when *n*-hexane was used as extraction solvent. A central composite design (CCD) was applied to select the most appreciated operating condition. The method performance was evaluated according to the SANTE/11945/2015 guidelines and was linear in the 0.1 to 200 µg/L range for 10 phthalate compounds, with *r*^2^ > 0.996 and individual residuals <15%. Repeatability (RSD_r_), within-laboratory reproducibility (RSD_wr_), and the trueness range were from 2.7 to 9.1%, from 3.4 to 14.3% and from 91.5 to 118.1%, respectively. The limit of detection (LOD) was between 0.5 to 1.0 ng/L and the limit of quantitation (LOQ) was between 1.5 to 3.0 ng/L for all 10 compounds. The developed method was successfully applied to the analysis of non-alcoholic beverages.

## 1. Introduction

Vietnam is a developing country with a high demand for soft drinks. According to the Vietnam beverage association (VBA) the average consumption of non-alcoholic beverages is over 43 L/person/year and the market is expected to develop from 4 billion liters in 2017 to an estimated 5 billion liters in 2020. With the positive growth of beverages market, consumers increasingly pay more attention to the quality of bottles along with the impact on human health due to long storage time and high temperature conditions.

Polyvinyl chloride (PVC) and polyethylene (PE) are main raw materials of plastic bottles that are widely used in a broad variety of beverages in many countries around the world [1]. In the manufacture process, phthalates are used to produce flexible and durable plastics. It is estimated that in 2017 the global consumption of phthalate compounds was nearly 8 million tons [2]. Because phthalates do not form chemical bonds in the plastic’s network, they are easily released and migrate into food and beverages at different stages such as packaging, bottling and production [3,4,5]. As a result, the consumers can inevitably be exposed to phthalates via eating and drinking. The presence of phthalates was detected in more than 95% of human urine samples in numerous countries around the world [6,7]. Bioaccumulative potential, toxicity and adverse effects of phthalates on experimented animals have been reported in previous studies [8,9]. Phthalates are known as endocrine disrupters, severely affecting respiratory, hepatic and reproductive organs [10,11,12,13].

In 1999, the United States Environmental Protection Agency (U.S. EPA) added 8 phthalates to the list of hazardous chemicals, of which di (2-ethylhexyl) phthalate (DEHP) was in the category of carcinogenic substances of level II [14]. The Chemicals Act (REACH) of the European Council have introduced three phthalate compounds including DEHP, di-*n*-butyl phthalate (DBP) and benzyl butyl phthalate (BzBP) into Section 52, Appendix XVII (Annex XVII of the REACH Regulation) since 2007 in order to restrain the production, trade and use of these compounds [15]. Since then, the list of prohibited phthalate substances has increased steadily. RoHS 2 EU/65/2011 amended in 2016 remains the restrictions of DEHP, BzBP, and DBP concentration <0.1% (enacted since 2011) and will be forbidden in all electrical and electronic devices as of 22/7/2019.

Currently, there has been no research and statistics in Vietnam about phthalates concentrations and the risk of exposure to consumers due to these compounds being contained in plastic bottles. To support consumers becoming more aware of the hazards of these toxins and selecting healthy food, the assessment of phthalates in types of non-alcoholic beverage drinks is vitally important.

Therefore, in this study, the presence of 10 phthalates in 148 samples of non-alcoholic beverages divided into 6 groups in Vietnam was investigated by liquid–liquid extraction and gas chromatography-tandem mass spectrometry (GC-MS/MS) [16]. We also applied response surface methodology (RSM) to plan the sample preparations.

## 2. Materials and Methods

### 2.1. Chemicals and Materials

Individual neat crystal phthalates standards, including Dimethyl phthalate (DMP), diethyl phthalate (DEP), dipropyl phthalate (DPP), diisobutyl phthalate (DiBP), benzyl butyl phthalate (BzBP), di-*n*-hexyl phthalate (DnHP), di (2-ethylhexyl) phthalate (DEHP), di-*n*-octyl phthalate (DnOP), dicyclohexyl phthalate (DCHP) and di-*n*-butyl phthalate (DBP) and three isotope titrants (dimethyl phthalate-3,4,5,6-d4 (DMP-d4), diisobutyl phthalate-3,4,5,6-d4 (DiBP-d4) and di (2-ethylhexyl) phthalate-3,4,5,6-d4 (DEHP-d4)) were obtained from Sigma (St. Louis, MO, USA). The purities of phthalate standards and isotope internal standards were guaranteed above 98%. Only glassware was used in all analytical procedure. All the containers such as volume flasks, centrifuge tubes, pipettes and extraction funnels are rinsed carefully by methanol, ethyl acetate and *n*-hexane. Standard stock solution of 10 phthalate compounds and the isotope titrants were prepared by dissolving each compound in *n*-hexane to obtain solutions with concentration of 1000 mg/L and then diluted to 10 mg/L. The internal isotope solutions were prepared at the same procedure to the concentration of 100 mg/L. Standard solutions containing 0.1, 1.0, 5.0, 10.0, 20.0, 50.0, 100.0 and 200.0 µg/L of the 10 phthalates in *n*-hexane were prepared daily and used for the preparation of calibration curves.

Methanol and acetonitrile were purchased from Thermo Fisher (Waltham, MA, USA). Dichloromethane, *n*-hexane, sodium sulfate, sodium chloride and sodium hydroxide were obtained from Merck (Waltham, MA, USA) with purity of above 95%. Ultrapure water was prepared by Milli-Q^®^ Gradient A10 (Merck Millipore, Burlington, MA, USA).

Fourteen commercial mineral water, 17 carbonated drinks, 29 functional drinks, 32 juice drinks, 33 tea drinks and 23 fermented milks were purchased from the retail market in Hanoi, Vietnam. All non-alcoholic beverages were assigned with unique marks and protected from the light until needed.

### 2.2. Instrumentation and Chromatographic Conditions

All the phthalates determination was performed using a GC-MS/MS system (Thermo Fisher Scientific, Waltham, MA, USA), a Trace GC 1310 gas chromatograph, a TriPlus RSH Autosampler and TSQ 8000 mass spectrometer (Thermo, Waltham, MA, USA) and controlled by a computer running TraceFinder software. A DB5-MS (30 m × 0.25 mm × 0.25 µm) gas chromatography column from Agilent (Santa Clara, CA, USA) was used to separate phthalates. Oven temperature was set initially at 100 °C (hold for 1 min), then increased to 280 °C at 10 °C/min and to 310 °C at 5 °C/min. At 310 °C, temperature was maintained for 5 min. Helium was used as a carrier gas in a constant flow of 1 mL/min and the injection volume was 1 µL with an autosampler in splitless mode. The total of analysis time was 20 min. Solvent delay was 1 min. The GC was interfaced by a heated transfer liner (310 °C) to the mass spectrometer in electron ionization mode with an electron energy of 70 eV. Inlet temperature was 290 °C and inject volume was 1 µL. The criteria for the identification of phthalates were based on both the same retention times as the standard within ±2% and correctly relative abundance of two characteristic ions within ±15%. Data processing was done by TraceFinder software from Thermo Fisher Scientific. Identifying and quantifying ions, retention time, and collision energy are listed in Appendix A.

### 2.3. Sample Preparation Procedure

We injected 5.00 mL of samples into a 15 mL centrifuge tube, and then added 10 μL of internal isotope and 1.5 mL of methanol. The mixtures were mixed well by vortex and transferred to the extraction funnel. Next, 15 mL *n*-hexane was added to the funnel, and the mixtures were shaken vigorously for 7 min. After standing for 5 min to separate phases, 0.5 mL of 10% NaCl solution was added to remove the emulsion. The *n*-hexane solvent phases were transferred to 50 mL centrifuge tube. The procedure was repeated one more time, then the solutions after 2 extractions were transferred into erlenmeyer flasks and mixed vigorously. Next, 15 g Na_2_SO_4_ was added and shaken seriously to remove water completely. The remaining solutions were evaporated to about 5 mL by a rotary evaporator, and then dried by nitrogen until dry. Finally, the dried samples were dissolved in 1 mL *n*-hexane, filtered through a 0.22 μm Polytetrafluoroethylene (PTFE) filter and analyzed by means of the GC-MS/MS.

### 2.4. Experimental Design

Several trials were conducted to optimize a liquid–liquid extraction process for the quantitative analysis of phthalates in soft drinks. The D-optimal was selected to confirm the significant variables (V_solvent_/V_sample_ ratio, number of extractions, NaCl concentration and extraction time). The response was the sum of all phthalate peak areas. Two different full factorial designs were created at three levels: low (−1), medium (0) and high (+1). MODDE 12.1 software was used to design experimental matrices, calculate regression values and analytical variance. The D-optimal planning method was used with three continuous variables and one intermittent variable. A total of 29 experiments were done and the experimental results are shown in Table 1.

The relationship between the response function Y and the coded variables (X1, X2, X3, and X4) is indicated in the following equation:Y = β_0_ + β_i_∑x_i_ + β_ii_∑x^2^_i_ + β_i j_∑x_i_x_j_(1)
where Y is a response function; x_i_ and x_j_ are independent variables; β_0_ is a constant; and β_i_, β_ii_, and β_ij_ are linear, quadratic, and interactive coefficients, respectively.

The appropriate fitting model for the response was selected based on the comparison of various statistical parameters such as *R*^2^, *Q*^2^, lack of fit and adequate precision.

### 2.5. Figure of Merit

Validating the analysis method in this research followed the instruction of European SANTE 11945/2015. The parameters evaluated in the validating process for carbonated beverages and fat drink samples are linearity, linear range, recovery, precision, limit of detection (LOD) and limit of quantitation (LOQ). Quantification was carried out by the internal calibration method. To assess the specificity, blank samples were tested based on the extraction process in order to evaluate false positive phenomenon and contamination of the chemicals. To appraise the linear range, 7 values of mix standard solution of 10 phthalate compounds with the concentration of 1.0, 5.0, 10.0, 20.0, 50.0, 100.0 and 200.0 µg/L were prepared. LOD is defined as the three times the standard deviation of eleven consecutive blank injections divided by the slope of the calibration curve (LOD = (3 × (SD_blank_) (slope of the calibration cure))) and LOQ is calculated based on the lowest spike level for which the criteria for trueness (i.e., 70–120%) and precision (<20%) met. Precision is calculated using 15 determinations (i.e., three concentration levels in quintuplicate). The repeatability (RSD_r_) is calculated from the results of four replicate experiments in a single day of standard 1, 10 and 100 µg/L and the within-laboratory reproducibility (RSD_wr_) is calculated from results obtained over four consecutive days. The trueness is calculated depending on the method of standard addition with the help of three different concentrations (1, 10 and 100 µg/L).

## 3. Results and Discussion

### 3.1. Selecting Extraction Solvent and Optimizing the Method of Solvent Evaporation

The analytical method of phthalates in beverage samples is based on the liquid–liquid extraction technique, and thus the recovery of the compounds relies on two fundamental factors: (i) removing solvent to extract compounds out of the matrix, and then eliminating the matrix and (ii) evaporating the solvent to concentrate the samples after extraction.

#### 3.1.1. Solvent Evaporation Method

One of the most common disadvantage of the liquid–liquid extraction technique is a high volume of extraction solvent. It is, therefore, required to have a method that evaporates only the solvent but not the analytical substance. Herein, we implemented the evaluation of three solvent evaporation methods: (1) using nitrogen to remove the solvent, (2) using the rotary evaporator system, and (3) combining these two methods. The standard solution was mixed in 30 mL *n*-hexane, which was carried out in experimental conditions as mentioned above. The result was reflected through the recovery of 10 phthalate compounds (Figure 1).

The recovered efficiency of 10 phthalates when using the vacuum rotary evaporator to evaporate solvent ranged from 12% to 62%; while using the nitrogen gas flow, the figure was from 32% to 72%. However, when combining these methods, the recoveries of all 10 phthalates were higher than single methods, ranging from 91% to 105%. This combination saved analysis time, and reduced the evaporation of substances as well as contact time between the substances and surrounding atmosphere. Therefore, we incorporated vacuum rotary evaporator into nitrogen gas to evaporate the solvent in the sample preparation process.

#### 3.1.2. Selecting the Extraction Solvent

The requirements of extraction solvents using in liquid-liquid extraction technique are dissolving well the analytical compounds, having strong affinity to the compounds and preventing matrix effect. In this study, we assessed the extraction ability of numerous solvents such as *n*-hexane, chloroform (CHCl_3_), dichloromethane (CH_2_Cl_2_) and ethyl acetate (CH_3_COOC_2_H_5_). The isotope standard solution was prepared and added to the drinks which contain fats. These mixtures and each of the mentioned solvents were evaporated by the combination of the rotary evaporator and nitrogen gas. The results of solvent selection were based on the sum of the chromatographic peak area of the three isotopes, and are shown in Figure 2.

As shown in Figure 2, the total peak area of three internal standard substances of using *n*-hexane as an extraction solvent was much higher than those of dichloromethane, chloroform and ethyl acetate. Although chloroform is a perfect candidate for extraction of many substances, in this case, when using this solvent in the extraction of beverage samples, other chemicals are also extracted into the organic phase leading to a decrease of the internal standard peak. Therefore, it cannot detect the signal of the internal standard peak. This problem is similar to that of dichloromethane and ethyl acetate solvents [17,18,19,20]. Nonetheless, the obtained area when using *n*-hexane was the highest one because this solvent has better capability to extract the analytical compounds to the organic phase and more relatively eliminates the matrix effect than utilizing dichloromethane. To sum up, we decided to use *n*-hexane as the extraction solvent in this study.

### 3.2. D-Optimal

Analysis of variance is widely used to predict the suitability of a model with experiment results. The obtained results (Table 2) indicated that the predicted values of the model were not conflict with the experiments. The coefficient of determination of *R*^2^ was 0.932 and the coefficient of determination adjustment *R*^2^_adj_ was 0.910. The suitability of the model was also shown in *P* values and Fisher test. *P*_regression_ value was 0.000 (<0.05), and *P*_Lack of fit_ was 0.221 (>0.005), which showed that the obtained model was consistent with the experiment.

The three-dimensional response surface shows the effect and interaction of the two factors on the target function. Figure 3a shows the combined effect of the V_solvent_/V_sample_ ratio and NaCl concentration. Figure 3b shows the image effect of NaCl concentration and time of extraction. Interaction between the V_solvent_/V_sample_ ratio and time of extraction is shown in Figure 3c. In general, when the value of the variables increases, the efficiency of the phthalate extraction rises and eventually reaches equilibrium.

The contribution of these factors on the extraction efficiency is shown in Figure 3d. The solvent/sample ratio was the biggest influence (54.8%), followed by extraction time (35.8%), and final NaCl concentration (9.4%).

The optimal tool of MODDE 12.1 software was used for the optimization. The results are shown in Table 3. Experimental result was obtained at optimum conditions, yield was 92.51 (95% confidence). This proves that the model was highly meaningful, allowing good experimental results.

### 3.3. Method Performance

To evaluate the efficiency of the proposed method, a number of parameters of the method were investigated and manifested in Table 4, Table 5 and Table 6. The triple quadrupole detector provided a high degree of selectivity. The linear ranges of these phthalate compounds were built up from 1 to 200 µg/L. Additionally, the weights 1/x^2^ shown through the correlation coefficient (*r*^2^) were greater than 0.996, a non-significant lack of fit and individual residuals deviation of <13% proved the quality of the method. The lowest LOD of these phthalate substances was 0.5 ng/L and the highest was 3.0 ng/L. The maximum of retention time was ±0.06 min, which was below the maximum tolerance deviation stated in SANTE guidelines (±0.1 min). The repeatability (RSD_r_) and within-laboratory reproducibility (RSD_wr_), which expressed percent relative standard deviation (%RSD), ranged from 1.0 to 9.1% and from 2.7 to 12.3%, respectively. All detected RSD values were smaller than 15% that meet the SANTE guideline of RSD ≤ 20%. The trueness of this method was appraised through the recovery values by adding standard solution to carbonated beverage and fat beverage samples at three different concentrations. The average recoveries of 10 phthalate compounds are demonstrated in Table 3 and are within the range required by the SANTE guidelines (between 70% and 120%).

### 3.4. Levels of Phthalates in Samples

Non-alcoholic drink samples were analyzed based on the above sample preparation method. The results are shown in Table 7 and Figure 4. As described in Table 4, DBP and DEHP were also detected in all of the 148 collected beverage samples which were analyzed, while DnOP were found in 33% of samples. The appearance of phthalate compounds ranged from 1% to 100% so that almost all of the samples were contaminated by phthalates. As can be seen in Figure 4, DEHP was the phthalate which primarily presents in the samples (>35%), followed by DBP and DEP in mineral water, fruit juice, tea, fermented milk and functional drink. Conversely, in carbonated drink samples, DnOP was the most abundant phthalate substance (>50%). In relation to fermented milk, DMP and DEHP were comparatively in the same proportion (45.5 and 47.6%). It was recognizable that there was the extensive appearance of DMP, DnOP and DBP. DnHP virtually did not appear in these kinds of beverage drinks. DnOP was chiefly found in carbonated drink samples (54.3%). Additionally, DMP was mainly detected in fermented milk samples (45.5%).

The concentrations of phthalate compounds presenting in non-alcoholic beverages are also illustrated in Table 8. In 6 groups of experimental samples, DEHP was the phthalate substance containing the highest mean as well as medium value among all of the samples. The mean and medium were 91.6 and 64.5 µg/L. The variation of concentrations of 148 samples ranged from 0.092 to 466.6 µg/L, which were much higher than those of DBP (22.1 and 18.8 µg/L, the variation varies from 0.093 to 73.5 µg/L). The DMP, BzBP, DPP. DiBP, DnOP and DCHP contents were nd–131.9 µg/L, 0.30–21.5 µg/L, nd –0.52 µg/L, nd–1.9 µg/L, nd–200.4 µg/L and nd–0.60 µg/L, respectively.

Table 8 also shows that the concentrations of phthalate compounds studied on different targets were considerably different. The DEHP in fruit juice samples had the highest mean (230.8 µg/L) and median (222.7 µg/L) among other types of beverage drinks in this experiment. Moreover, the mean and median of DEP (17.9 and 17.3 µg/L) in fruit juice drink were also far higher than other beverages. In terms of fermented milk, DMP content was detected in a range of 12.3 to 131.9 µg/L, and the average and median were 68.0 and 65.7 µg/L, respectively.

The distribution of the total phthalate concentration in non-alcoholic drinks was different among the sample matrices. Juice drinks had the highest phthalate concentration, followed by fermented milk and tea. As shown in Figure 5, DEHP was a major contributor leading to the phthalate contamination in non-alcoholic beverage, similar to the previous study [21]. The contamination of phthalates depended on the characteristics of the samples. The sample containing preservatives (potassium benzoate) had higher phthalate concentration than that which did not use preservatives [22]. Furthermore, the sample carrying high fat content was easier to contaminate by phthalate [23]. When comparing the data above, the identification of the sources of phthalate contamination was ambiguous because of other factors such as temperature, pH, light, turbidity and storage time [5,24,25].

Among all phthalates, DEHP is the most popular substance appearing in non-alcoholic beverages in similar studies. Figure 6 illustrates the degree of DEHP contamination in recent studies and the container of these products is not necessarily made from plastic. According to the research of Ustun et al., DEHP concentrations of soda, lemonade, mineral water and high-taste water in Turkey ranged from 73 to 2312 ng/g and the highest DEHP concentration was found in Cola soft drink [21]. In contrast, based on the study of Sireli et al., DEHP concentration in fruit juice drink varies from 1.1 to 44.3 ng/g, which is much lower than Ustun’ research [26]. Wu et al. reported that the DEHP content in energy drink and tea ranged from 15 to 83 ng/g [27]. DEHP concentration was remarkably high in the study of Truong et al. of chocolate and high-fat drinks (111–1753 ng/g) [23]. In our research, DEHP concentration varied from 0.1 to 466.6 ng/g, remained within the range of the above studies and predominantly concentrated in milk-containing fruit juice sample.

### 3.5. Exposure to Phthalates

Assessing phthalate concentration in non-alcoholic beverages has been investigated by many researchers around the world. However, in Vietnam, there are no specific statistics on phthalate content in daily beverage drinks. Identification of the existence as well as frequency of the occurrence of phthalate compounds in the matrices totally depends on instrument detection limit (IDL) and method detection limit (MDL) of the study, but comparison of phthalate contamination in non-alcoholic beverages still has scientific meaning.

Relying on the studies of Guo [28] and Sireli [26], we calculated the daily intake of DEP, DBP, BzBP and DEHP in Vietnam following the formula below:(2)$$\mathrm{EDI}=\frac{\mathrm{CQ}}{\mathrm{bw}}r_{\mathrm{uptake}}$$
where EDI (µg/kg × day) is the estimated daily intake from drinking beverages, C (ng/g) is the phthalate concentration in beverages, r is the gastrointestinal uptake factor and bw (kg) is the body weight. In this study, average beverages intake was 150 g/day, r_uptake_ was 1 and an average bw of 50 kg was used for Vietnam population. The result is shown in the Table 9.

The daily intake of DEHP when investigating phthalates in beverages in Vietnam was higher than TDI (U.S. EPA), but the contamination of DEP, DBP and BzBP was significantly lower than the threshold of regulation. The phthalate concentration in non-alcoholic beverages did not give rise to serious consequences for adult health. However, the beverages such as fruit juice and fermented milk, which were analyzed, are consumed daily by pregnant women. Because of this, there is likely to be a mother-to-child exposure through the placenta [29] leading to the phenomenon of hormonal disturbance in children [30].

## 4. Conclusions

In this research, we focused on the assessment of phthalate compounds in beverages, products whose consumption has grown dramatically in Vietnam, and thus the phthalate contamination factor for non-alcoholic drink was not exactly reflected the exposure level. Phthalate exposure in daily life possibly originates from different sources such as air [31,32,33], food [28,34], beverages [35,36] as well as cosmetics [37]. As a consequence, in this study, we solely concentrated on evaluating phthalate in beverage drinks, products which are consumed in huge quantities in Vietnam.

Liquid–liquid extraction and the GC-MS/MS analysis technique were optimized and conducted successfully in determining 10 phthalate compounds in different kinds of non-alcoholic drinks. The good recoveries (70–120%), RSDs of all the analysis samples and matrices were lower than 15% and low LOQ (0.5 ng/L) was confirmed. This method was utilized to analyze 10 phthalate substances in 148 non-alcoholic drink samples. The result showed that 100% of the samples were contaminated by DEHP and DEP, and almost all samples were polluted by phthalates. The result of phthalate contamination in this study did not reflect accurately the exposure of phthalates in beverage drinks because of other influencing factors. Therefore, it is necessary to implement more in-depth research to assess properly phthalate contamination during the production process, storage conditions, and when the human body is exposed to these products.

## Figures and Tables

**Figure 1 toxics-06-00069-f001:**
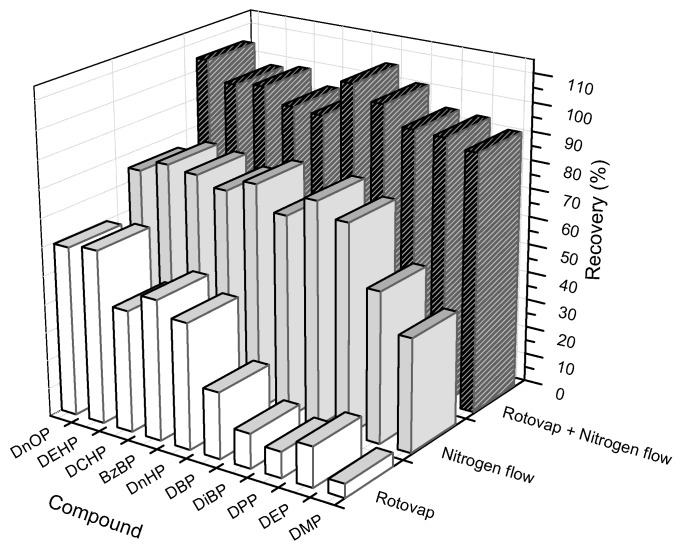
Comparison of recoveries between variable solvent evaporation methods. Note: Dimethyl phthalate, DMP; Diethyl phthalate, DEP; Dipropyl phthalate, DPP; Diisobutyl phthalate, DiDP; Benzyl butyl phthalate, BzBP; di-*n*-hexyl phthalate, DnHP; di(2-ethylhexyl) phthalate, DEHP; di-*n*-octyl phthalate, DnOP; Dicyclohexyl phthalate, DCHP and di-*n*-butyl phthalate, DPBP.

**Figure 2 toxics-06-00069-f002:**
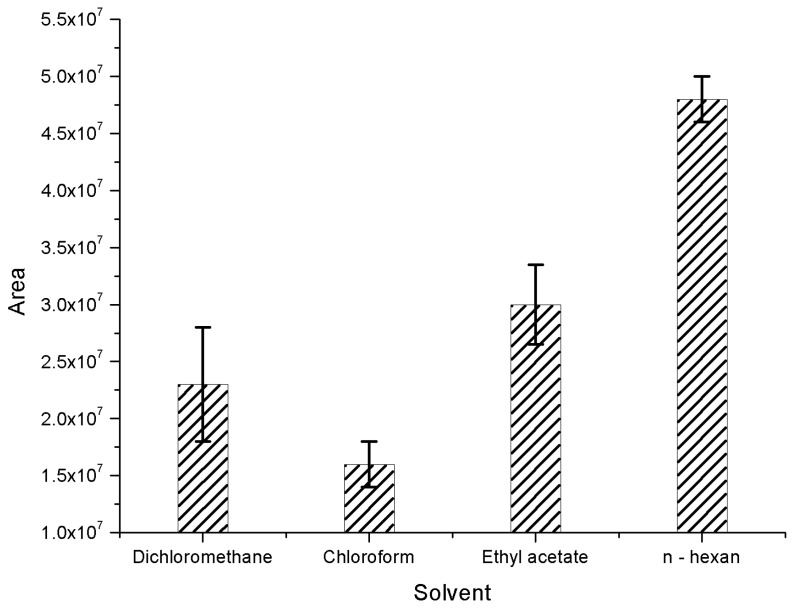
Total peak area of 3 isotope internal standards when using different solvents.

**Figure 3 toxics-06-00069-f003:**
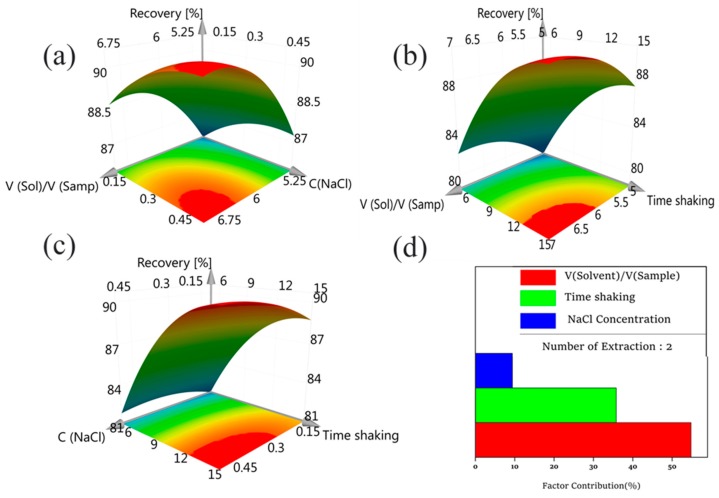
Response surface plots for the central composite design (CCD). Note: (**a**) V_solvent_/V_sample_ ratio vs. NaCl concentration; (**b**) NaCl concentration vs. time of extraction; (**c**) V_solvent_/V_sample_ ratio vs. time of extraction and (**d**) The phthalate extraction rises vs. eventually reaches equilibrium.

**Figure 4 toxics-06-00069-f004:**
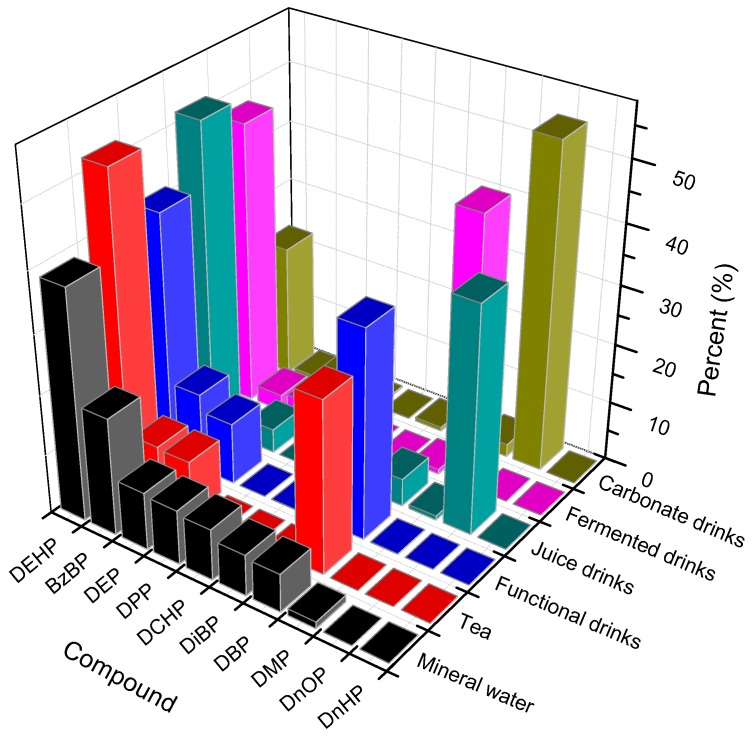
The distribution of 10 phthalate compounds in different kinds of samples. Note: Dimethyl phthalate, DMP; Diethyl phthalate, DEP; Dipropyl phthalate, DPP; Diisobutyl phthalate, DiDP; Benzyl butyl phthalate, BzBP; di-*n*-hexyl phthalate, DnHP; di (2-ethylhexyl) phthalate, DEHP; di-*n*-octyl phthalate, DnOP; Dicyclohexyl phthalate, DCHP and di-*n*-butyl phthalate, DPBP.

**Figure 5 toxics-06-00069-f005:**
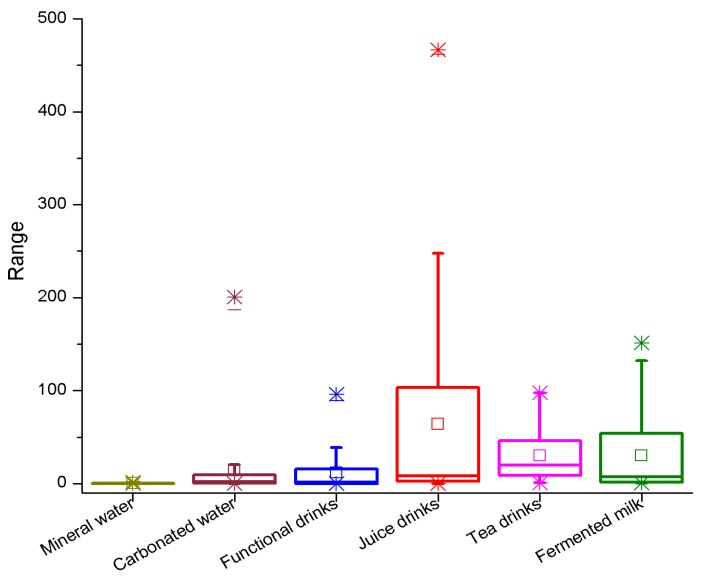
The distribution of the total phthalate concentration in non-alcoholic drinks.

**Figure 6 toxics-06-00069-f006:**
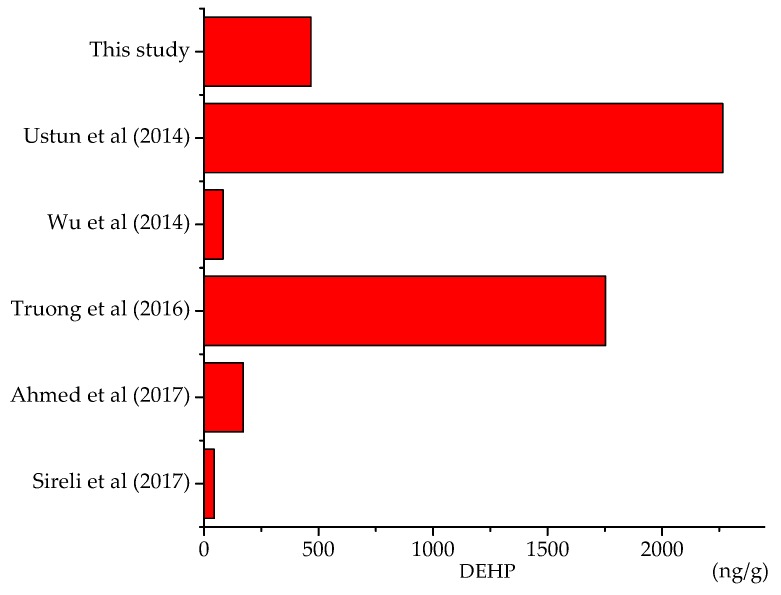
Concentrations of DEHP in similar studies.

**Table 1 toxics-06-00069-t001:** Independent variable and experiment level.

Coded	Independent Variable	Level Experiment		
		−1	0	1
X1	V_solvent_/V_sample_ ratio	5.0	6.0	7.0
X2	Number of extractions	1	2	3
X3	NaCl (M)	0.1	0.2	0.3
X4	Time of extraction	5	10	15

**Table 2 toxics-06-00069-t002:** Analysis of variance (ANOVA).

Recovery	DF	SS	MS (Variance)	F	*P*	SD
Total corrected	28	1199.89	42.8532			6.54624
Regression	7	1118.59	159.798	41.2734	**0.000**	12.6411
Residual	21	81.3055	3.87169			1.96766
Lack of Fit	19	79.1989	4.16836	3.9573	**0.221**	2.04166
(Model error)						
Pure error	2	2.10667	1.05333			1.02632
(Replicate error)						
	N = 29	*Q*^2^ =	0.857	Cond. no. =	1.752	
	DF = 21	*R*^2^ =	0.932	RSD =	1.968	
	Comp. = 2	R^2^_adj._ =	0.910			

Note: degrees of freedom, DF; sum of squares, SS; mean square, MS; Fisher, F; probability value, P; and standard deviation, SD.

**Table 3 toxics-06-00069-t003:** Optimization of phthalate extraction process.

V_solvent_/Vs_ample_ Ratio	Number of Extractions	NaCl Concentration (M)	Solvent	Drying Method	Time of Extraction (Min)	% Recovery Efficiency	
						Predicted	Experiment
6.5	2	0.42	*n*-hexan	Rotovap + nitrogen flow	14	90.7	91.1

**Table 4 toxics-06-00069-t004:** Linear dynamic range (µg/L), determination coefficients (*r*^2^), residuals, retention times, limit of detection (LOD) and limit quantitation (LOQ).

Compound	Linear Dynamic Range (µg/L)	*r* ^2^	Maximum Individual Residual (%)	Retention Times (min)	LOD (ng/L)	LOQ (ng/L)
DMP	0.1–200	0.997	10.3	6.58	1.0	3.0
DEP	0.1–200	0.996	11.2	8.16	1.0	3.0
DPP	0.1–200	0.998	9.8	10.12	1.0	3.0
DiBP	0.1–200	0.999	11.5	12.00	0.5	1.5
BzDP	0.1–200	0.997	7.3	15.52	1.0	3.0
DnHP	0.1–200	0.997	12.9	15.42	1.5	4.5
DEHP	0.1–200	0.999	9.1	17.02	0.5	1.5
DnOP	0.1–200	0.999	8.2	18.42	1.0	3.0
DCHP	0.1–200	0.996	10.2	16.90	1.0	3.0
DBP	0.1–200	0.999	11.1	12.33	1.0	3.0

**Table 5 toxics-06-00069-t005:** Repeatability (RSD_r_) and within-laboratory reproducibility (RSD_wr_) for peak areas evaluated at three concentration levels.

Compound	RSDr (*n* = 5)			RSDwr (*n* = 5 × 4 Days)		
	1 µg/L	10 µg/L	100 µg/L	1 µg/L	10 µg/L	100 µg/L
DMP	3.6	1.0	3.2	9.1	4.1	5.1
DEP	6.3	4.3	3.8	6.7	4.7	3.4
DPP	7.5	5.4	4.3	8.5	3.3	3.8
DiBP	2.6	1.7	5.7	14.3	3.3	3.3
BzDP	1.6	2.8	3.6	8.6	2.7	3.3
DnHP	5.6	6.0	3.0	12.2	2.8	2.7
DEHP	2.4	3.7	4.4	10.2	7.0	2.8
DnOP	3.8	5.7	8.2	11.2	6.7	4.1
DCHP	5.2	7.4	9.1	12.3	8.6	4.7
DBP	1.9	2.3	8.8	9.9	6.7	3.3

data is presented as % RSD.

**Table 6 toxics-06-00069-t006:** Trueness results for 10 phthalate compounds in non-alcoholic beverages matrices.

Spiking Level Compound	1		10		100	
	Gas (M/R)	Fat (M/R)	Gas (M/R)	Fat (M/R)	Gas (M/R)	Fat (M/R)
DMP	109.2/7.3	103.2/3.4	94.9/4.3	100.6/9.1	99.2/3.8	103.3/3.8
DEP	97.2/3.8	92.6/6.3	98.3/1.0	98.9/8.8	108.3/1.0	109.3/5.7
DPP	102.2/4.2	101.2/7.5	91.5/1.9	101.4/2.3	94.0/1.7	103.1/5.6
DiBP	107.1/9.1	102.8/2.6	92.3/5.2	105.2/7.4	102.2/4.3	114.7/10.1
BzDP	99.6/6.9	99.3/1.0	113.3/3.8	101.3/5.7	108.5/1.9	97.5/1.6
DnHP	105.6/3.6	100.5/1.9	111.1/2.4	96.5/3.7	100.2/2.8	108.9/8.8
DEHP	114.7/10.1	105.1/2.4	101.1/5.6	99.7/6.0	101.1/5.7	103.2/7.5
DnOP	110.5/8.6	102.5/3.7	99.5/1.6	1002/2.8	94.3/5.2	118.1/12.3
DCHP	117.6/6.9	103.0/1.7	102.8/2.6	103.0/1.7	107.7/6.0	96.6/6.3
DBP	118.1/12.3	109.2/2.8	103.2/7.5	96.1/5.4	100.6/3.6	105.4/8.2

Note: M: Mean (% recovery); R: Relative standard deviation (%).

**Table 7 toxics-06-00069-t007:** Detection of frequency [n (%)] of phthalates in non-alcoholic beverages in Hanoi.

Non-Alcoholic Beverages	N	DMP	DEP	DBP	BzBP	DEHP	DnHP	DPP	DiBP	DnOP	DCHP
Mineral water	14	14 (100)	14 (100)	14 (100)	14 (100)	14 (100)	0	14 (100)	14 (100)	0	14 (100)
Carbonated drinks	17	14 (82)	17 (100)	17 (100)	16 (94)	17 (100)	0	17 (100)	17 (100)	17 (100)	17 (100)
Functional drinks	29	19 (66)	26 (90)	29 (100)	28 (97)	29 (100)	0	25 (86)	29 (100)	0	29 (100)
Juice drinks	32	31 (97)	32 (100)	32 (100)	32 (100)	32 (100)	0	32 (100)	0	0	0
Tea drinks	33	0	33 (100)	33 (100)	33 (100)	33 (100)	0	0	0	0	0
Fermented milks	23	23 (100)	23 (100)	23 (100)	23 (100)	23 (100)	0	0	0	0	0
Total	148	101 (68)	145 (98)	148 (100)	146 (99)	148 (100)	0	88 (59)	60 (41)	49 (33)	60 (41)

**Table 8 toxics-06-00069-t008:** Phthalate concentrations in different types of non-alcoholic beverages (µg/L).

Non-Alcoholic Beverages		Mineral Water	Carbonated Drinks	Functional Drinks	Juice Drinks	Tea Drinks	Fermented Milk
DMP	Range	0.02–0.05	1.1–5.5	0.04–0.08	0.1–5.6	Nd	12.3–131.9
	Mean	0.03	3.3	0.06	3.3		68
	Median	0.04	3.4	0.05	3.2		65.7
DEP	Range	0.05–0.57	0.38–13.0	1.6–17.9	3.0–33.4	0.9–21.1	0.6–10.2
	Mean	0.3	4.8	10	17.9	11.2	5
	Median	0.29	3.4	11	17.3	10.3	3.8
DBP	Range	0.09–0.95	0.18–43.5	0.16–64.7	1.6–73.4	6.0–4.3	0.81–1.35
	Mean	0.61	19.8	30.3	20.6	34.9	1.1
	Median	0.59	19	26.3	17.8	34.7	1.1
BzBP	Range	0.30–0.95	0.31–3.4	0.41–21.5	0.4–6.4	0.8–21.9	1.1–8.4
	Mean	0.61	1.4	11.8	3.1	11.5	4.2
	Median	0.59	1.2	11.7	3	11.3	3.7
DEHP	Range	0.46–1.8	0.09–71.0	0.72–96.1	27.7–466.6	15.4–87.8	8.3–151.2
	Mean	1.2	28.6	35.9	230.8	63.4	71.2
	Median	1.2	20.5	21	222.7	72.9	67.6
DnHP	Range	Nd	Nd	Nd	Nd	Nd	Nd
	Mean						
	Median						
DPP	Range	0.13–0.44	0.05–0.52	Nd–0.06	0.03–0.21	Nd	Nd
	Mean	0.28	0.3	0.01	0.11		
	Median	0.28	0.33	0.01	0.1		
DiBP	Range	0.14–0.41	0.04–1.9	Nd–0.14	Nd	Nd	Nd
	Mean	0.22	0.76	0.07			
	Median	0.15	0.83	0.06			
DnOP	Range	Nd	0.98–200.4	Nd	136–197	Nd	Nd
	Mean		69.2		169.1		
	Median		56.9		169		
DCHP	Range	0.13–0.60	0.02–0.09	Nd–0.15	Nd	Nd	Nd
	Mean	0.26	0.07	0.02			
	Median	0.16	0.07	0.01			

**Table 9 toxics-06-00069-t009:** Characteristics of the investigated phthalates.

Compound	EDI (µg/kg × Day)	TDI (µg/kg × Day)
DEP	1.50 × 10^−4^–0.1002	0.800 (U.S. EPA)
DBP	2.70 × 10^−4^–0.2202	0.100 (U.S. EPA)
BzBP	9.00 × 10^−4^–0.0645	0.800 (U.S. EPA)
DEHP	2.70 × 10^−4^–1.3998	0.020 (U.S. EPA)

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
