# Peer review of "Optimization of Sample Preparation for Detection of 10 Phthalates in Non-Alcoholic Beverages in Northern Vietnam"

_toxics, 2018, doi:10.3390/toxics6040069_

Round 1

Reviewer 1 Report

The work "optimization of sample preparation for detecting of ten phthalates in non-alcoholic beverages in Northern ........"as a whole is good scientific work and brings interesting applications for determination of phthalates, in my opinion to be published in Toxics the authors must cover the English language and update the references. I suggest the authors to consider the following works:

1) Fast determination of phthalate ester residues in soft drinks and light......; MV Russo, I Noradonato, P Avino, G Cinelli. RCS Advances 4(103), 59655-59663(2014)

2) Extraction and GC-MS analysis of phthalate esters in food matrices: a review. MV Russo, P Avino, L Perugini, I Notardonato; RSC Advances 5 (46) 37023-37043 (2015)

Author Response

Point:

The work "optimization of sample preparation for detecting of ten phthalates in non-alcoholic beverages in Northern ........"as a whole is good scientific work and brings interesting applications for determination of phthalates, in my opinion to be published in Toxics the authors must cover the English language and update the references. I suggest the authors to consider the following works:

1) Fast determination of phthalate ester residues in soft drinks and light......; MV Russo, I Noradonato, P Avino, G Cinelli. RCS Advances 4(103), 59655-59663(2014)

2) Extraction and GC-MS analysis of phthalate esters in food matrices: a review. MV Russo, P Avino, L Perugini, I Notardonato; RSC Advances 5 (46) 37023-37043 (2015)

Respond:

Thank you very much for your comment and suggestion. We revised our manuscript and updated the reference. Two suggested references were added into our work.

Reviewer 2 Report

In this manuscript, the authors present a comprehensive study of phthalates determination in non-alcoholic beverages in Vietnam, using liquid-liquid extraction with GC-MS/MS.  The sample preparation method was specifically optimized and the analysis method was well evaluated by investigating the related parameters.  The manuscript is well organized.  I recommend it to be published after following points are addressed.

It would be better if the authors could provide more information and details about the tandem mass spec. experiment. 

The first paragraph in section 2.2 should be removed, which is duplicate from the second paragraph in section 2.1. 

Some Figures and Tables are misnumbered in this manuscript, which need to be fixed.

Author Response

Thank you very much for your comments

Point 1:

It would be better if the authors could provide more information and details about the tandem mass spec. experiment. 

Respond 1:

We updated these information in the revised text and tables

Point 2:

The first paragraph in section 2.2 should be removed, which is duplicate from the second paragraph in section 2.1. 

Respond 2:

We removed the paragraph in section 2.2

Point 3:

Some Figures and Tables are misnumbered in this manuscript, which need to be fixed.

Respond 3:

We revised our manuscript with updated information.
